# CHST7 Methylation Status Related to the Proliferation and Differentiation of Pituitary Adenomas

**DOI:** 10.3390/cells11152400

**Published:** 2022-08-04

**Authors:** Wei Dong, Wenjian Shi, Yongliang Liu, Jingwu Li, Yu Zhang, Guilan Dong, Xiaoliu Dong, Hua Gao

**Affiliations:** 1Department of Neurosurgery, The People’s Hospital of Tangshan County, Tangshan 063000, China; 2Department of Tumor Surgery, Tangshan People’s Hospital, Tangshan 063000, China; 3Department of Intensive Care Medicine, Tangshan People’s Hospital, Tangshan 063000, China; 4Department of Oncology, Tangshan People’s Hospital, Tangshan 063000, China; 5Department of Neurology, Tangshan People’s Hospital, Tangshan 063000, China; 6Beijing Neurosurgical Institute, Capital Medical University, Beijing 100050, China

**Keywords:** pituitary adenomas, lineage, CHST7, cell proliferation, cell differentiation, methylation, CHST71, pituitary adenomas, DNA methylation, tumor differentiation, tumor proliferation

## Abstract

Pituitary adenomas (PAs) are the second most common primary brain tumor and may develop from any of the cell lineages responsible for producing the different pituitary hormones. DNA methylation is one of the essential epigenetic mechanisms in cancers, including PAs. In this study, we measured the expression profile and promoter methylation status of carbohydrate sulfotransferase 7 (CHST7) in patients with PA; then, we investigated the effect of the CHST7 methylation status on the proliferation and differentiation of PAs. The volcano map and Metascape results showed that the levels of CHST7 were related to the lineages’ differentiation and the cell adhesion of PAs, and patients with low CHST7 had greater chances of having an SF-1 lineage (*p* = 0.002) and optic chiasm compression (*p* = 0.007). Reactome pathway analysis revealed that most of the DEGs involved in the regulation of TP53 regulated the transcription of cell cycle genes (HSA-6791312 and HSA6804116) in patients with high CHST7. Correlation analysis showed that CHST7 was significantly correlated with the eIF2/ATF4 pathway and mitochondrion-related genes. The AUC of ROC showed that CHST7 (0.288; 95% CI: 0.187–0.388) was superior to SF-1 (0.555; 95% CI: 0.440–0.671) and inferior to FSHB (0.804; 95% CI: 0.704–0.903) in forecasting the SF-1 lineage (*p* < 0.001). The SF-1 lineage showed a higher methylation frequency for CHST7 than the Pit-1 and TBX19 lineages (*p* = 0.009). Furthermore, as the key molecule of the hypothalamic–pituitary–gonadal axis, inhibin βE (INHBE) was positively correlated with the levels of CHST7 (r = 0.685, *p* < 0.001). In summary, CHST7 is a novel pituitary gland specific protein in SF-1 lineage adenomas with a potential role in gonadotroph cell proliferation and lineage differentiation in PAs.

## 1. Introduction

Pituitary adenomas (PAs) are the second most common type of primary brain tumor, affecting up to 5% of the general population, with serious complications [1]. Pituitary adenomas may develop from any of the cell lineages and hypersecrete hormones or cause mass effects, which include headaches, hypopituitarism, and visual field defects. The initial therapy is generally transsphenoidal surgery, with medical therapy being reserved for those not cured by surgery, except for PiT-1-lineage tumors and particularly SF-1-lineage tumors. There is currently no guidance or expert consensus on the medical treatment of the SF-1 lineage, although there are several clinical trials, including ClinicalTrials.gov NCT0327191 and ClinicalTrials.gov NCT00939523 [2,3]. In recent years, the concept of refractory Pas was proposed for those Pas characterized by a high Ki-67 index, rapid growth, frequent recurrence, and resistance to conventional treatments [4]. However, the relationship between tumor differentiation, receptor status, and hormone production is not fully understood. Therefore, it is important to gain a comprehensive understanding of the molecular biological characteristics of SF-1-lineage tumors.

Glycobiology plays the key role in various biological and medical fields, and the accumulation of chondroitin sulfate proteoglycans (CSPGs) plays a key role in tumorigenesis by driving the expression of oncogenes and promoting their crucial interaction in the tumor microenvironment [5]. The activity-dependent remodeling of CSPGs’ main extracellular matrix (ECM) could be involved in the structural plasticity of the hypothalamo-neurohypophysial system [6]. Carbohydrate sulfotransferase 7 (CHST7) transfers the sulfate to carbohydrate groups in glycoproteins and involved in carcinogenesis by controlling the biosynthesis of CSPGs [7]. The CHST7 promoter hypermethylation increased the risk of colorectal cancer. Serum CHST7 will be used for the differentiation between lung cancer and non-malignant pulmonary inflammation [7]. The level of CHST7 was noticeably altered in the breast epithelial cell line MCF-10F after parathion treatment, which has a synergistic effect with estrogen [8]. 

It is known that the majority of Pas are sporadic, with no high-frequency driver mutation. Epigenetic modifications can greatly influence tumorigenesis and tumor differentiation in subtypes of Pas [9]. Cell-lineage-specific transcription factors play the key role in the development of pituitary cells [10]. The methylation-directed glycosylation of chromatin factors may be a possible mechanism regulating retrotransposon promoters [11]. In this study, we measured the expression and methylation status of the CHST7 promoter in PA patients, compared among three cell lineages. We then analyzed the relationship of the CHST7 expression profile with PA tumor proliferation, invasion, and differentiation. 

## 2. Materials and Methods

### 2.1. Patient Samples

All the subjects were collected at the People’s Hospital of TangShan from March 2014 to April 2015. There were 106 patients, including 72 males and 34 females, with a mean age of 44.2 ± 5.7 years (range, 21–75). There were 32 cavernous sinus compression cases and 37 fast-proliferative cases (Ki-67 index > 3%). The population included 20 Pit-1-lineage cases, 15 T-PIT-lineage cases, and 71 SF-1-lineage cases, which were diagnosed. The study protocols conformed to the ethical guidelines of the Declaration of Helsinki (RMYY-LLKS-2020-004) and were approved by the Internal Review Board of the People’s Hospital of TangShan. 

### 2.2. Quantitative Methylation-Specific PCR

DNA (15–50 mg) was extracted from 30 adenoma specimens using TRIzol reagent according to the manufacturer’s protocol (Invitrogen, CA, USA). The quantity and quality of DNA were measured according to the values at 260/280 nm (Nanodrop ND-1000, ILN, USA). Bisulfite treatment was performed using the EZ-96 DNA methylation kit (Zymo Research, Orange, CA, USA). The DMR positions and methylation probes of CHST7 gene were obtained from HM450. The methylation status of CHST7 was measured according to MS-HRM assay (range: chrX: 46433054-46434902). Quantitative methylation-specific PCR was performed by real-time PCR. The adjusted methylation level was calculated by dividing the raw methylation level by the tumor cell fraction in a sample.

### 2.3. Differential Gene Enrichment Analysis and Gene Set Enrichment Analysis (GSEA)

LIMMA package was used for the differential analysis. GO and KEGG enrichment analysis was measured according to the clusterProfiler package and enrichplot package (https://github.com/GuangchuangYu/enrichplot, accessd on 28 August 2021) using the Metascape database. GSEA was performed for the prognostic role of CHST 7 (http://software.broadinstitute.org/gsea/index.jsp, accessd on 1 March 2022) [12]. GSEA uncovered the potential mechanisms of prognostic CHST7 genes using the Molecular Signatures Database (MSigDB, http://software.broadinstitute.org/gsea/msigdb/index.jsp) c2, c5, and c7. The false discovery rates (FDRs) < 0.25, |Normalized Enrichment Scores (NESs)| > 1, and nominal *p*-values < 0.05 were considered to indicate statistically significant differences.

### 2.4. Quantitative Reverse Transcription PCR (RT–PCR)

Twenty-four adenoma specimens (~10 mg) were extracted the total RNA using the RNeasy^®^ Mini Kit (Qiagen, Düsseldorf, Germany). RT-qPCR was performed as described previously using the Applied Bio-systems 7500 Fast System (Life Technologies, Carlsbad, CA, USA) [13]. The fold-change of difference was calculated using the comparative CT method. 

### 2.5. Immunohistochemistry (IHC)

The tissue microarray was constructed as described previously. The microarrays were cut into 4 μm sections and incubated with anti-CHST7 (rabbit polyclonal, 1:500, ab154726, Abcam), anti-Pit-1 (rabbit monoclonal, 1:1,000, ab273048), anti-T-PIT (mouse monoclonal, 1:750, ab243028, Abcam, Cambridge, UK), anti-SF1 (rabbit monoclonal, 1:1000, ab217317), and anti-Ki-67 (rabbit monoclonal, 1:400, ab16667) primary antibodies. BOND Polymer Refine Detection (Leica Biosystems, DS9800, Weztlar, Germany) was used to detect the primary antibodies. The H-score was obtained by multiplying the staining intensity by a constant to adjust the mean to the strongest intensity to yield a score ranging from 0 to 300.

### 2.6. SDS-PAGE and Western Blot Analyses

A total of 30 μg/lane protein was loaded onto 10% Bis-Tris SDS-PAGE gels, separated electrophoretically, and blotted onto polyvinylidene fluoride membranes (Merk, New York, NY, USA). The blots were incubated with antibodies against anti-CHST7 (ab154726, 1:2000), anti-EIF2B5 (ab91565, 1:1000), anti-EIF2D (ab108218, 1:2000), anti-EIF2AK4 (ab32384, 1:4000), and anti-LC3 I/II (14600-1-AP, Proteintech, Chicago, IL, USA), followed by a secondary antibody (1:8000) tagged with horseradish peroxidase (Santa Cruz Biotechnology, Dallas, TX, USA). The blots were visualized by enhanced chemiluminescence, and densitometry was performed using a fluorescent image analyzer (Amersham Imager 600, GE, Chicago, IL, USA). 

### 2.7. Statistical Analysis

All the data are expressed as the means ± standard errors. Student’s *t*-test was used for normally distributed continuous data, and the Mann–Whitney U test for non-normally distributed continuous data. The marker combinations model was constructed by logistic regression, and the diagnostic performance was evaluated by the receiver operating curve (ROC). The correlation was evaluated by Pearson’s chi-square or Fisher’s exact test. *p*-values < 0.05 were considered statistically significant. The SPSS software version 17.0 (IBM Corp., Armonk, NY, USA) was used for the statistical analyses. 

## 3. Results

### 3.1. Tumor and Patient Characteristics

Transcription factors are essential for the differentiation and maturation of neuroendocrine cells, which include three main cell lineages: the pituitary-specific POU-class homeodomain transcription factor (POU1F1, also known Pit-1) lineage, the steroidogenic factor 1 (SF-1) lineage, and the T-box family member TBX19 (T-PIT) lineage. In this cohort, there were 20 Pit-1-lineage patients, 15 T-PIT-lineage patients, and 71 SF-1-lineage patients (Table 1). There were more female patients with the T-PIT lineage compared to Pit1 or SF-1 (χ^2^ = 10.62, *p* = 0.005). There was also a statistically significant difference in age among the three lineages (43 ± 2.67 vs. 50.65 ± 2.29 vs. 54.7 ± 2.1 years, F(2, 103) = 4.927, *p* < 0.001). There were no statistical differences in tumor proliferation or tumor invasion among the three lineage adenomas (*p* > 0.05). 

### 3.2. CHST7 Methylation Status in Different Lineages of PA Patients

In this study, 106 samples had sufficient tissue with a tumor cell content ≥70% available for the methylation experiment. CHST7 hypermethylation was defined as a methylation mean ≥50%, while <20% was defined as hypomethylation. We determined that the CHST7 methylation status of 33/106 (31.1%) cases was hypermethylation; for 43/106 (40.6%), it was medium, and for 30/116 (28.3%) cases, it was hypomethylation (Table 2). Compared to the medium and hypomethylation groups, there were more fast proliferative cases in the hypermethylation group (*p* = 0.026) and more optic chiasm compression (*p* = 0.001). The tumor volume was 11.6 ± 3.86 cm^3^ in the hypermethylation group, 11.69 ± 3.59 cm^3^ in the medium group, and 15.67 ± 4.91 cm^3^ in the hypomethylation group (*p* = 0.742). In addition, there were no statistical differences in sex and invasive behavior, such as cavernous sinus compression and skull destruction, among the different methylation statuses of the CHST7 promoter. 

We found three hypermethylation, seven medium, and ten hypomethylation statuses in the Pit-1-lineage patients; one, eight, and six in the T-PIT-lineage patients; and twenty-nine, twenty-eight, and fourteen in the SF-1-lineage patients, respectively (χ^2^ = 13.39, *p* = 0.009). We did not observe any difference in age (51.97 ± 1.32 vs. 48.49 ± 1.61 vs. 52.37 ± 2.11, *p* = 0.063) according to CHST17 methylation status.

### 3.3. Differentially Expressed Gene (DEG) Enrichment Analysis between the CHST7 Groups

According to the mRNA expression of CHST7, the patients were divided into a high-CHST group and a low-CHST group. Based on the different levels of CHST7, 3606 gene sets were significant at FDR < 25%, and 1107 gene sets were significantly enriched at a nominal *p* value < 1%. The volcano map of the characteristic differences based on pairwise comparisons is shown (Figure 1A), and it includes the characteristic molecules of the SF-1 lineage: FSHB, NR5A1 (also named SF-1), and TGFBR3L [14]. Immunologic signature gene sets, including dendritic cells, regulated T cells, and perturbations of CD4+/CD8+ cells were significantly enriched in the low-CHST group (Figure 1B). DEG pathway enrichment using the Metascape database showed that the top three pathways were responses to metal ions (R = HSA-5660526), FOXO-mediated transcription (R = HSA-9614085), and TP53-regulated transcription of cell cycle genes (R = HSA-6791312). The differential genes in the low-CHST group were closely related to organ development, cell adhesion, and the regulation of hormone levels, as shown in Figure 1C. 

ER and nutrient stress promote the assembly of respiratory chain supercomplexes through the eIF2/ATF4 axis [15]. In this study, we found the correlation coefficients of CHST and EIF2AK3 (r = 0.672, *p* < 0.001) was the highest compared to EIF2AK1 or EIF2AK2 in Figure 2A, and the correlation coefficients of CHST and EIF2B2 (r = 0.614, *p* < 0.001) was the highest compared to EIF2B1 or EIF2B3 in Figure 2B. 

Correlation analysis showed the Spearman correlation coefficients (r) of CHST7 and POU1F1 to be 0.762 (*p* < 0.001), SSTR2 to be 0.683 (*p* < 0.001), and DLL3 to be 0.918 (*p* < 0.001) in Appendix A.

Furthermore, we analyzed the correlation of CHST7 and mitochondrion-related genes, and the correlation coefficient of CHST7 and LAMP1 was 0.569 (*p* < 0.001) and that of OPA1 was 0.534 (*p* < 0.001) in Appendix A. 

Based on the correlation coefficients of the genes, we chose POU1F1 and DLL3 to assess the tumor differentiation and EIF2AK3 to assess the eIF2/ATF4 pathway for further verification in eight patients by western blot experiment in Figure 3A. Western blot experiments with the clinical samples also demonstrated that CHST7 protein expression was correlated with EIF2AK3, DLL3, and POU1F1 (r = −0.412, r = 0.446, and r = 0.446, *p* < 0.05, respectively) (Figure 3B). 

### 3.4. The CHST7 Methylation Status Relates to the Cell Differentiation of PAs

The RT-PCR experiments showed that the mRNA levels of POU1F1 for the hypomethylation group were the highest, followed by those for the medium group and the hypermethylation group (Figure 4A) (*p* = 0.032). There was no statistical difference in the Pit-1 level between the medium and hypermethylation groups (*p* > 0.05). The mRNA levels of TBX19 with hypermethylation were the lowest, and there was no statistical difference in the TBX19 level between the methylation and hypermethylation groups (Figure 4B) (*p* = 0.122). Interestingly, there were statistically significant differences in the SF-1 levels between the three groups of patients (Figure 4C) (*p* < 0.001). We found that the levels of ESR1/2 and SSTR2/5 in the hypermethylation group were the lowest, and they were the highest in the hypomethylation group (Figure 4D–G) (*p* < 0.05). These results provide the possibility of a clinical trial with an estrogen receptor antagonist and somatostatin analog in the SF-1 lineage, especially in CHST7-promoter-hypomethylation patients. The protein levels of transcription factors and receptors from the IHC experiment are shown (Appendix A). 

The high-CHST17-expressing patients included 10 with T-PIT, 16 with Pit-1, and 27 with SF-1 lineages compared to 5, 4, and 44, respectively, among the low-CHST17 patients, as shown in Table 3 (χ^2^ = 12.94, *p* = 0.002). Higher Pit-1 and SSTR2 expression was observed in the high-CHST17 patients compared to low-CHST17 patients (*p* < 0.01), and there was no statistical difference in TBX19 or SF-1 between the two groups (Figure 5A–G). The correlation coefficient of the CHST7 ΔCT values and Pit-1 ΔCT was 0.343 (95% CI: 0.158 to 0.505, *p* < 0.001) (Figure 6A), and that between CHST7 and SSTR2 was 0.657 (*p* < 0.001) (Figure 6B). An ROC curve was drawn to determine the possibility for distinguishing the lineages of PA. The result showed that the area under the ROC curve (AUC) of CHST7 was 0.699 for the Pit-1 lineage (Figure 6C), 0.675 for the T-PIT lineage (Figure 6D), and 0.269 for the SF-1 lineage (Figure 6E). In the SF-1 lineage, the ROC value of CHST7 was better than that of either ESR1 or SF-1, which were adopted as classification indices for gonadotroph adenoma in the 2017 WHO classification. 

### 3.5. The Level of CHST7 Is Related to the Proliferation of PAs

To explore the role of CHST7 in PAs, we measured the mRNA levels of CHST7 and analyzed the relationship of CHST7 with clinical features. The low-CHST7 group had a higher proliferative index than the high-CHST7-patient group (Ki67 > 3%: 30/53 vs. 7/53, *p* < 0.001), as shown in Table 3. There was more optic chiasm compression (27/53 vs. 11/53, *p* = 0.001) and more impaired vision (24/53 vs. 11/53, *p* = 0.007) observed in the low-CHST7 group than in the high group. There were no statistical differences in age (51.02 ± 1.6 years vs. 51.96 ± 1.3 years), sex (male/female: 37/16 vs. 35/18), tumor size (15.85 ± 3.96 cm^3^ vs. 10.04 ± 2.44 cm^3^), cavernous sinus compression, or skull destruction between the two groups (*p* > 0.05). In addition, we did not find a statistical difference in the genes related to epithelial–mesenchymal transition and angiogenesis. 

### 3.6. Inhibin βE (INHBE) mRNA Expression Was Positively Correlated with CHST7

Inhibins/activins are involved in regulating a number of diverse functions of the hypothalamic–pituitary–gonadal (HPG) axis [16]. As a novel pituitary-gland-specific protein, TGFBR3L is an inhibin B co-receptor that regulates the follicle-stimulating hormone and female fertility [14,17]. In this study, we noticed that there was a positive correlation between CHST7 and INHBC (r = 0.337, *p* = 0.018) and INHBE (r = 0.428, *p* = 0.002), as shown in Figure 7A,B. We observed a higher level of INHBE mRNA in the high-CHST7 patients than in the low-CHST7 patients (33/53 vs. 20/53, *p* = 0.012). ANOVA showed that there was a statistically significant difference in the INHBE level between the three lineages. The highest INHBE expression was in the Pit-1 lineage, and the lowest was in the SF-1 lineage (F = 10.03, *p* < 0.001) (Figure 7C).

The 106 patients were divided into a high-INHBE and low-INHBE group, according to the median value of INHBE, as shown in Table 4. The high-IHNBE patients included 15/15 cases from the T-PIT lineage, 16/20 from the Pit-1 lineage, and 22/71 from the SF-1 lineage (χ^2^ = 32.47, *p* = 0.000). The high-INHBE patients in the hypomethylation group included 1/1 case in the T-PIT lineage, 2/3 cases in the Pit-1 lineage, and 5/29 cases in the SF-1 lineage; for the methylation group, there were 8/8 cases, 6/7 cases, and 10/28 cases, respectively; there were 6/6, 8/10, and 7/7 in the hypermethylation group, respectively (χ^2^ = 14.14, *p* = 0.001). In this study, the ROC value of INHBE was 0.851 (95% CI: 0.777–0.924, *p* = 0.000) in the T-PIT lineage, 0.139 (95% CI: 0.067-0.211, *p* = 0.000) in the SF-1 lineage, and 0.744 (95% CI: 0.631–0.856, *p* = 0.001) in the Pit-1 lineage, as shown in Figure 7D–F.

## 4. Discussion

Although histopathology is the first-line approach for the diagnosis of PAs, hormone immunohistochemistry and pituitary transcription factor markers currently enable more accurate characterization of tumors in this field. In this study, we found that CHST7 hypermethylation in PAs was associated with the tumor proliferation and cell differentiation of SF-1-lineage adenomas, and CHST7 hypomethylation was related to the cell differentiation of the Pit-1 and T-PIT lineages. RT-PCR and IHC experiments further showed that low CHST7 or its loss may contribute to the inhibition of Pit-1 and TBX19 and the activation of SF-1. In addition, differential gene enrichment analysis showed that CHST7 was involved in the regulation of the eIF2/ATF4 pathway and mitochondrion-related genes. 

Chondroitin sulfates are linear glycosaminoglycans containing repeating disaccharide units of glucuronic acid (GlcUA) and N-acetyl-D-galactosamine (GalNAc) [18]. Chondroitin sulfate can bind to various cytokines and growth factors, cell surface receptors, adhesion molecules, enzymes, and fibrillar glycoproteins of the extracellular matrix, thereby influencing both the cell behavior and the biomechanical and biochemical properties of the matrix [19]. The removal of chondroitin sulfate leads to altered associations of N-cadherin-positive neural progenitors and reduces Sox2 expression [20]. In the breast cancer cell line BT-549, chondroitin sulfates enhanced the invasive ability through inducing the proteolytic cleavage of N-cadherin [20]. The targeted silencing of Rb by proteoglycan neuron-glial antigen 2 resulted in PAs with immunohistochemical and ultrastructural features that resembled those of aggressive Pit1-lineage tumors [21].

In this study, we found that low-CHST7 patients had a high Ki67 index and a greater chance of having impaired vision than high-CHST7 patients, and there were no statistically significant differences in tumor behaviors, such as cavernous sinus compression or skull destruction, between the two groups. There were 36.7% with cavernous sinus invasion. Nearly 26% were shown by histological proof in the Pit-1-lineage adenomas [22], 41.3% in the SF-1-lineage adenomas [23], and 43% in the T-PIT-lineage adenomas [24]. We speculated that CHST7 inhibited tumor proliferation but not cavernous sinus invasion based on the RT-PCR experiments. 

Tumor cells are addicted to elevated protein synthesis for the augmented activity of most components of the translation system [25]. Translation initiation lies downstream of the deregulated signaling pathways in cancer and is regulated by activated oncogenes or mutated tumor suppressors. Therefore, controlling protein synthesis would have significant potential for exploring innovative therapy in somatotroph adenomas. In this study, we found a significant difference in the EIF2/ATP4 pathway based on different levels of CHST7 according to GSEA. We also proved a positive correlation between the CHST7 and EIF2/2ATF4-pathway-related genes, including EIF2B5, EIF2D, EIF2AK4, and EIF2AK3. Immunologic signature gene sets showed that the level of CHST7 was related to dendritic cells, regulated T cells, and perturbations of CD4+/CD8+ cells. EIF2AK4 drives a shift in the phenotype of tumor-associated macrophages and myeloid-derived suppressor cells that promotes antitumor immunity [26]. Reciprocal regulation between EIF2AK3 and EIF2AK4 through the JNK–FOXO3 axis modulates cancer drug resistance and clonal survival [27]. 

At present, dopamine agonists and somatostatin analogs are used for NFPA patients. There have been some clinical trials of drugs meant to reduce tumor volume in the case of failure of surgery [2,28]. However, when pituitary hormone immunostaining in NFPAs is weak, dubious, or completely negative, the analysis of transcription factors is critical for the determination of pituitary-cell-lineage differentiation. SF-1 is a transcription factor related to gonadotroph differentiation. In our cohort, we found that nearly 20% of patients expressed the mRNA for SF-1/Pit-1 or SF-1/TBX19. Our data showed the same phenomenon of the co-expression of two transcription factors as previously described [29,30]. In this study, we found that CHST7 hypermethylation was related to SF-1-lineage differentiation, and CHST7 hypomethylation was related to Pit-1-lineage and T-PIT-lineage differentiation, which suggested a key role for CHST7 in the lineage differentiation of PAs. The RT-PCR and IHC experiments also showed that there were more patients with low CHST7 with SF-1-lineage adenomas and more patients with high CHST7 with a Pit-1 lineage. We found a positive correlation between the mRNA levels of CHST7, Pit-1, and SSTR2, which suggests that CHST7 may function at the crossroads of tumor cell differentiation. ROC curve analysis revealed that the mRNA levels of CHST7 and FSHB showed excellent diagnostic value for gonadotrophs rather than the mRNA level of SF-1. 

The member of TGF-β family, activin and inhibin, play the critical role in modulating the pituitary gonadotropes and the reproductive axis [31]. The activin could activate the FSGB promoter in gonadotrope cells and regulate the sensitivity to GnRH according to activating the GnRHR promoter and altering receptor expression [32]. As inhibitors of activin-dependent FSH expression and secretion, inhibins play a crucial feedback function in the reproductive axis. Inhibin was first identified as a gonadal hormone that potently inhibits pituitary FSH synthesis and secretion. In this study, we found a low INHBE level in SF-1-lineage patients and a high INHBE level in Pit-1- and T-PIT-lineage patients. Combined with the correlation analysis, we speculate that INHBE is a potential target gene of CHST7 involved in gonadotroph cell differentiation. 

Overall, we observed a positive relationship between CHST7 methylation and lineage differentiation in PAs. We speculated that CHST7 was involved in cell differentiation in PA. The identification of biological mechanisms and potential clinical applications should lead to important improvements in anti-PA treatments. Increasing knowledge about cell differentiation should provide the basis for an effective treatment strategy in PA patients.

## Figures and Tables

**Figure 1 cells-11-02400-f001:**
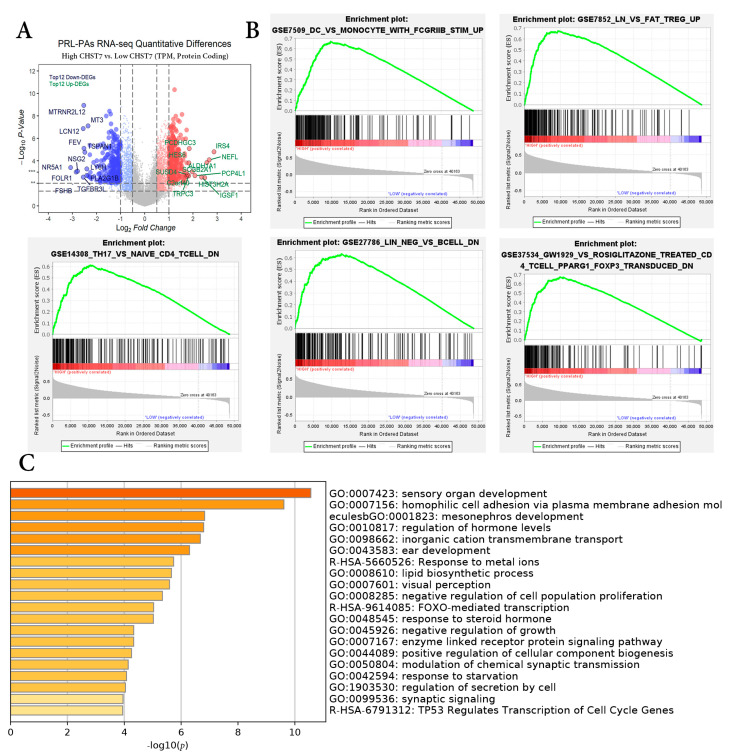
Differential gene enrichment analysis between different CHST7 groups. (**A**) Volcano plots showing the significantly differentially expressed genes. (**B**) GSEA employed to verify the signatures of the top 5 immunologic signature gene sets. (**C**) MetaScape enrichment analysis including pathways and GO terms.

**Figure 2 cells-11-02400-f002:**
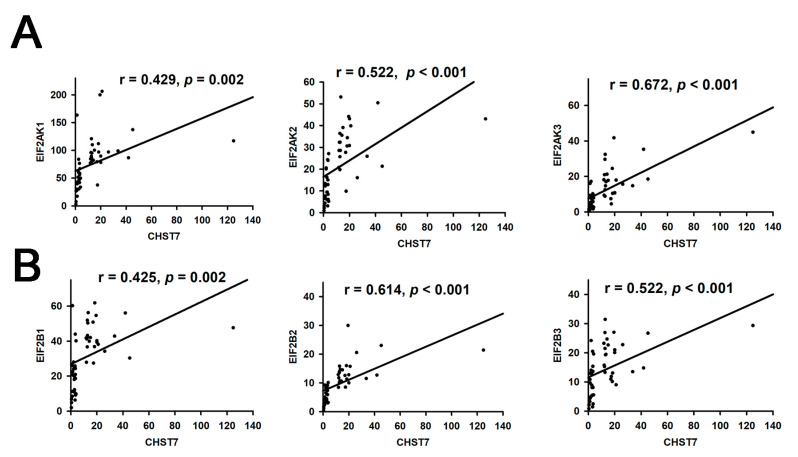
Correlation analysis of CHST7 and EIF2 pathway. (**A**) Correlation of CHST7 and EIF2AK1-3. (**B**) Correlation of CHSTY and EIF2B1-3.

**Figure 3 cells-11-02400-f003:**
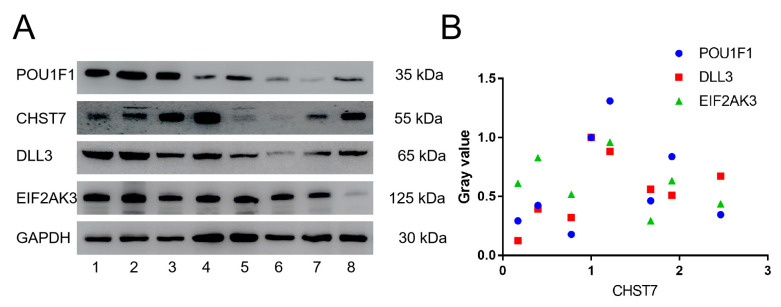
Western blot experiments of CHST7, POU1F1, DLL3, and EIF2AK3. (**A**) Electrophoretic band of CHST7, POU1F1, DLL3, and EIF2AK3 in 8 patients. (**B**) Correlation analysis of CHST7 and POU1F1, DLL3, and EIF2AK3. In this study, the gray value of patient 1 as 1.

**Figure 4 cells-11-02400-f004:**
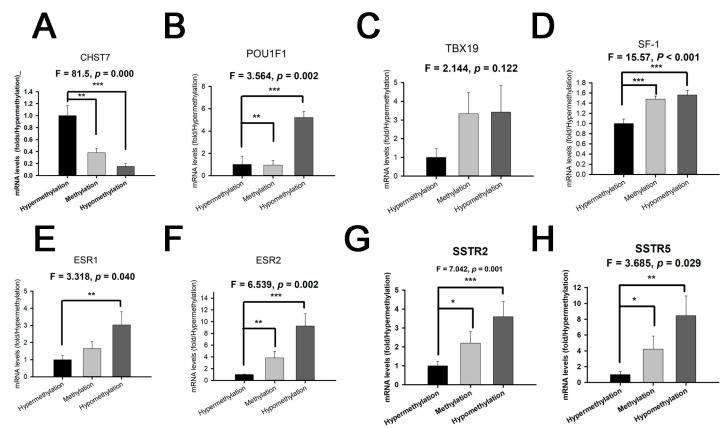
The mRNA levels of CHST7, transcription factors, and receptors grouped by different CHST7-promoter-methylation statuses. (**A**) CHST7. (**B**) Pit-1. (**C**) TBX19. (**D**) SF-1. (**E**) ESR1. (**F**) ESR2. (**G**) SSTR2. (**H**) SSTR5. Compared to the hypermethylation group: * *p* < 0.05; ** *p* < 0.01; *** *p* < 0.001.

**Figure 5 cells-11-02400-f005:**
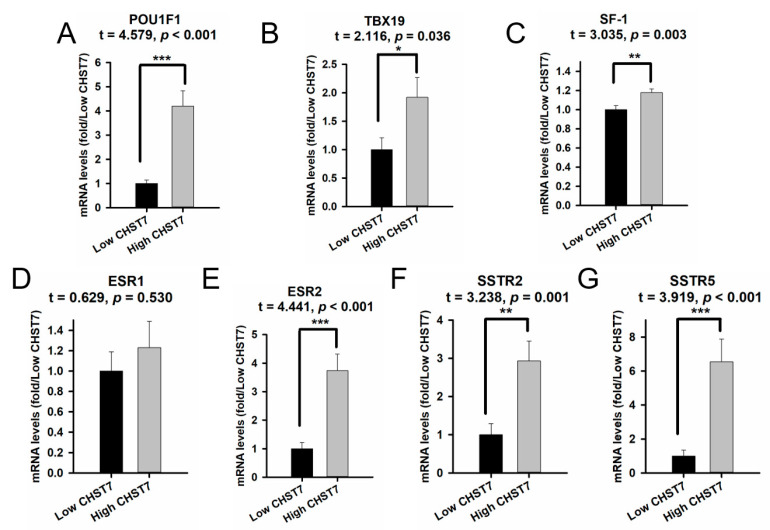
The mRNA levels of transcription factors and receptors compared with different mRNA levels of CHST7. (**A**) Pit-1; (**B**) TBX19; (**C**) SF-1; (**D**) ESR1; (**E**) ESR2; (**F**) SSTR2; (**G**) SSTR5. Compared to the low-CHST7 group: * *p* < 0.05; ** *p* < 0.01; *** *p* < 0.001.

**Figure 6 cells-11-02400-f006:**
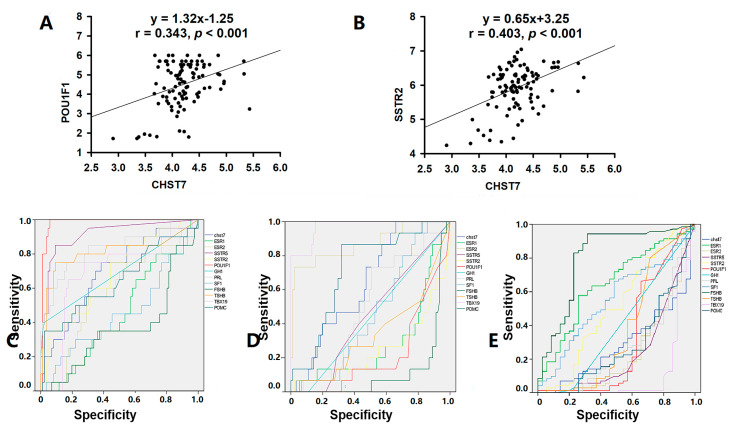
CHST7 is related to the tumor differentiation of PAs. (**A**) Correlation analysis of CHST7 and Pit-1 in 106 samples. (**B**) Correlation analysis of CHST7 and SSTR2 in 106 samples. (**C**) ROC curve of CHST7 for the Pit-1 lineage. (**D**) ROC curve of CHST7 for the TBX19 lineage. (**E**) ROC curve of CHST7 for the SF-1 lineage.

**Figure 7 cells-11-02400-f007:**
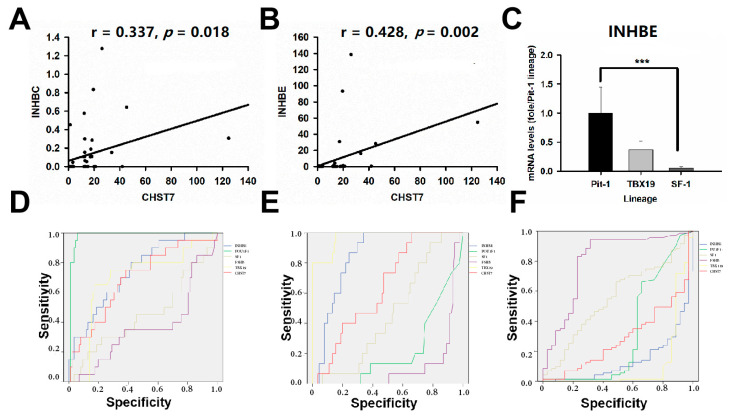
INHBE is related to the tumor differentiation of PAs. (**A**) The correlation analysis of CHST7 and INHBC. (**B**) The correlation analysis of CHST7 and INHBE. (**C**) mRNA levels of INHBE in the three lineages. *** *p* < 0.001. (**D**) ROC curve of INHBE for the Pit-1 lineage. (**E**) ROC curve of INHBE for the TBX19 lineage. (**F**) ROC curve of INHBE for the SF-1 lineage.

**Table 1 cells-11-02400-t001:** The clinical characteristics in 106 patients.

Variable	Lineage			*p* Value
	Pit-1	T-PIT	SF-1	
Sex				0.005
Male	16	5	52	
Female	4	10	19	
Age	50.65 ± 2.29	43.87 ± 2.67	54.7 ± 2.10	<0.001
Tumor volume (cm^3^)	9.34 ± 2.81	20.22 ± 9.61	12.2 ± 2.75	0.39
Ki67 index				0.411
>3%	5	7	25	
≤3%	15	8	46	
Skull destruction				0.901
Yes	4	3	17	
No	16	12	54	
Cavernous sinus compression				0.054
Yes	6	8	16	
No	14	7	55	
Optic chiasm compression				0.183
Yes	10	3	25	
No	10	12	46	

**Table 2 cells-11-02400-t002:** Correlation between CHST7 methylation and clinicopathologic features in 106 patients.

Variable	CHST7			*p* Value
	Hypomethylation	Medium	Hypermethylation	
Sex				0.449
Male	23	27	23	
Female	7	16	10	
Age	52.37 ± 2.11	48.49 ± 1.61	51.97 ± 1.32	0.063
Tumor volume (cm^3^)	15.67 ± 4.91	11.69 ± 3.59	11.6 ± 3.86	0.742
Ki67 index				0.026
>3%	6	14	17	
≤3%	24	29	16	
Skull destruction				0.273
Yes	4	10	10	
No	26	33	23	
Cavernous sinus compression				0.412
Yes	8	15	7	
No	22	28	26	
Optic chiasm compression				0.001
Yes	4	15	19	0.001
No	26	28	14	
Lineage				0.009
SF-1	14	28	29	
Pit-1	6	8	1	
T-PIT	10	7	3	

**Table 3 cells-11-02400-t003:** Association between CHST7 expression and clinicopathological characteristics.

Variable	CHST7		*p* Value
	Low	High	
Sex			0.677
Male	35	37	
Female	18	16	
Age	51.96 ± 1.3	51.02 ± 1.6	0.455
Tumor volume (cm^3^)	10.04 ± 2.44	15.85 ± 3.96	0.854
Ki67 index			0
>3%	30	7	
≤3%	23	46	
Skull destruction			0.225
Yes	14	10	
No	39	43	
Cavernous sinus compression			0.388
Yes	17	13	
No	36	40	
Optic chiasm compression			0.007
Yes	27	11	
No	26	42	
Lineage			0.002
SF-1	44	27	
Pit-1	4	16	
TBX19	5	10	

**Table 4 cells-11-02400-t004:** Association between INHBE expression and clinicopathological characteristics.

Variable	INHBE		*p* Value
	Low	High	
Sex			0.212
Male	39	33	
Female	14	20	
Age	53.08 ± 1.48	49.91 ± 1.42	0.125
Tumor volume (cm^3^)	10.19 ± 2.38	15.7 ± 4	0.241
Ki67 index			0.154
>3%	22	15	
≤3%	31	38	
Skull destruction			0.063
Yes	8	16	
No	45	27	
Cavernous sinus compression			0.196
Yes	12	18	
No	41	35	
Optic chiasm compression			0.105
Yes	23	15	
No	20	38	
Lineage			0.000
SF-1	49	22	
Pit-1	4	16	
TBX19	0	15	
CHST7 promoter			0.000
Hypermethylation	10	23	
Methylation	19	24	
Hypomethylation	24	6	

## Data Availability

The data presented in this study are available on request from the corresponding author.

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
