# Peer review of "CHST7 Methylation Status Related to the Proliferation and Differentiation of Pituitary Adenomas"

_cells, 2022, doi:10.3390/cells11152400_

Round 1

Reviewer 1 Report

The manuscript by Wei et al. presents methylation status of CHST7 gene, encoding carbohydrate sulfotransferase 7, in patients with pituitary adenoma (PA). The role of CHST7 in cancer biology is very limited, and this is the first study documenting any abnormalities in CHST7 status in PA patients. While the results presented are novel, the manuscript should be thoroughly revised.

Major comments:

  1. The paragraph in lines 197-205 is unclear and has to be re-written; grammar error in those sentences should be fixed, but it is also important what was the reason for the analysis of the correlation between CHST7 and those particular genes? The authors should clearly state what was the rationale for such a choice? Moreover, the results of the western blot analysis (mentioned at the end of this paragraph) should be described in much more detailed way; what is the difference between each of the 8 bands in each row? Such a description should be attached to the picture of the blot, but it should be also explained in the results section; why those 3 particular proteins were analyzed, while the other, encoded by the rest of the genes shown in Fig. 2A were not?
  2. The paragraph in lines 215-227 is not coherent with Fig. 3, e.g.: POU1F1 is missing in Fig. 3A, instead there is CHST7 in this figure (which is most likely a mistake); Fig. 3B presents Pit-1, but not TBX19, which is presented in Fig. 3C. If the difference between methylation and hypermethylation groups is not significant, than it is also not significant between hypermethylation and hypomethylation group (it should be stated in the text). SF-1 level is presented in Fig. 3D, not 3C.
  3. The subtitle 3.6. is unclear: which results show that INHBE is a target gene of CHST7? The correlation between the expression of two genes does not allow to state that one of them is a target of a protein encoded by another one.
  4. The titles of Table 3 and 4 are misleading. It is most likely the matter of grammar error, but they should be re-written.
  5. The conclusion concerning the involvement of CHST7 in the regulation of eIF2/ATF4 pathway is too far-fetched; please keep in mind that gene enrichment analysis is a powerful tool, but it provides some observation, not a mechanism(s); to get some mechanistic insights into the putative regulation of any pathway by CHST7 (or any other protein), several additional experiment are required; the sentences concerning this issue should be re-written with particular caution.

Minor:

The language should be improved.

Author Response

Major comments:

  1. The paragraph in lines 197-205 is unclear and has to be re-written; grammar error in those sentences should be fixed, but it is also important what was the reason for the analysis of the correlation between CHST7 and those particular genes? The authors should clearly state what was the rationale for such a choice? Moreover, the results of the western blot analysis (mentioned at the end of this paragraph) should be described in much more detailed way; what is the difference between each of the 8 bands in each row? Such a description should be attached to the picture of the blot, but it should be also explained in the results section; why those 3 particular proteins were analyzed, while the other, encoded by the rest of the genes shown in Fig. 2A were not?

Response:

Thanks for the question. We have re-written the sentences (line 197-205) and re-edited the language according to the native editor.

We also added the description of WB experiment as follows:

Based on the correlation coefficients of genes, w chose POU1F1 and DLL3 to assess the tumor differentiation, and EIF2AK3 to assess the eIF2/ATF4 pathway for further verification. And western blot experiment with clinical samples also demonstrated that CHST7 protein expression was correlated with EIF2AK3, DLL3 and POU1F1 (r=-0.412, r=0.446, r=0.446, p<0.05). 

  1. The paragraph in lines 215-227 is not coherent with Fig. 3, e.g.: POU1F1 is missing in Fig. 3A, instead there is CHST7 in this figure (which is most likely a mistake); Fig. 3B presents Pit-1, but not TBX19, which is presented in Fig. 3C. If the difference between methylation and hypermethylation groups is not significant, than it is also not significant between hypermethylation and hypomethylation group (it should be stated in the text). SF-1 level is presented in Fig. 3D, not 3C.

Response:

Thanks for the question.

We apologize for the writing errors. In fact, Pit-1 was the protein symbol and POU1F1 was the gene symbol. POU1F1 should be used in RT-PCR experiment. We added the CHST7 result in Figure 3, and not in text in the previous manuscript.

We have corrected the errors in revised the manuscript.

  1. The subtitle 3.6. is unclear: which results show that INHBE is a target gene of CHST7? The correlation between the expression of two genes does not allow to state that one of them is a target of a protein encoded by another one.

Response:

Thanks you for the question. We have re-edited the sentence as follows:

Inhibin βE (INHBE) mRNA expression was positively correlated to CHST7 involving gonodotroph cell differentiation.

  1. The titles of Table 3 and 4 are misleading. It is most likely the matter of grammar error, but they should be re-written.

Response:

Thanks you for the question. We have re-edited the titles of table3 and table 4 in revised the manuscript as following:

Table3. Association between CHST7 expression and clinico-pathological characteristics

Table4. Association between INHBE expression and clinico-pathological characteristics

  1. The conclusion concerning the involvement of CHST7 in the regulation of eIF2/ATF4 pathway is too far-fetched; please keep in mind that gene enrichment analysis is a powerful tool, but it provides some observation, not a mechanism(s); to get some mechanistic insights into the putative regulation of any pathway by CHST7 (or any other protein), several additional experiment are required; the sentences concerning this issue should be re-written with particular caution.

Response:

Thanks for the question. We have deleted the words according to the reviewer’s suggestion.

Minor:

The language should be improved.

Response:

Thanks for the question. We have re-edited the language according to the native editor.

Reviewer 2 Report

The manuscript by Dong Wei et colleagues explores the effects of CHST7 methylation status on proliferation and differentiation of pituitary adenomas with the aim to identify new biological mechanisms and potential clinical applications for PAs. The work is well conducted, and the authors' suggestions are well supported by the experiments. However the article need some revisions that should be addressed prior to the publication:

  • Authors demonstrated the involvement of eIF2/ATF4 pathways and the modulation of some mitochondria related genes; however, as the phosphorylation of eIF2 plays a pivotal role in the regulation of this pathway, I suggest the authors to assess (by Western blot fo example) also the expression levels of its phosphorylated form.
  • eIF/ATF4 involved other molecules related to UPR stress, to analyze the activated mechanism I suggest the authors to evaluate alterate expression of BiP and/or PERK or to better discuss their involvement in the discussion session.
  • In Fig. 2 the Western blot graph lack in clarity. Authors should indicate what the presented bands represent in the image. For each protein analyzed there are seven samples, which of them are non-tumor or tumor? Please, more detailed information are also needed in the figure legend and in the methods session.
  • Capitol letters indicating the graphs have different sizes in some figures. Furthermore, some graphs lack in quality (es. Fig.1D and 2A), please check if it is due to the draft version or provide better definition figures.
  • In Fig. 6D please add Sensitivity on y-axis legend

Author Response

  1. Authors demonstrated the involvement of eIF2/ATF4 pathways and the modulation of some mitochondria related genes; however, as the phosphorylation of eIF2 plays a pivotal role in the regulation of this pathway, I suggest the authors to assess (by Western blot fo example) also the expression levels of its phosphorylated form.

eIF/ATF4 involved other molecules related to UPR stress, to analyze the activated mechanism I suggest the authors to evaluate alterate expression of BiP and/or PERK or to better discuss their involvement in the discussion session.

Response:

Thanks for the question.

As Phosphorylation of the translation initiation factor eIF2 alpha at a conserved serine residue mediates translational control at the integrated stress response (ISR). We have realized the key role of EIF2/ATF4 pathway in the PAs differentiation. In our other draft, we found that the impaired amount of eIF2 subunits was related to the disease phenotype. IHC showed that the eIF2β protein is highly expressed in POU1F1 lineage adenomas compared with other subtypes. The knockdown of eIF2β also relieved the phenotype of POU1F1 lineage adenomas, including the declination of GH/IGF-1 level, improvement of acromegaly and contain of tumor proliferation. We would assess the expression levels of its phosphorylated form according to the reviewer’s suggestion.

  1. In Fig. 2 the Western blot graph lack in clarity. Authors should indicate what the presented bands represent in the image. For each protein analyzed there are seven samples, which of them are non-tumor or tumor? Please, more detailed information are also needed in the figure legend and in the methods session.

Response:

Thanks for the question.

We also added the description of the WB experiment as follows:

Based on the correlation coefficients of genes, w chose POU1F1 and DLL3 to assess the tumor differentiation, and EIF2AK3 to assess the eIF2/ATF4 pathway for further verification. And western blot experiment with clinical samples also demonstrated that CHST7 protein expression was correlated with EIF2AK3, DLL3 and POU1F1 (r=-0.412, r=0.446, r=0.446, p<0.05). 

  1. Capitol letters indicating the graphs have different sizes in some figures. Furthermore, some graphs lack in quality (es. Fig.1D and 2A), please check if it is due to the draft version or provide better definition figures.

Response:

Thanks for the question. We have re-edited the Figures in the revised manuscript according to the reviewer’s suggestion.

  1. In Fig. 6D please add Sensitivity on y-axis legend

Response:

Thanks for the question. We have re-edited the Figure 6D in revised manuscript according to the reviewer’s suggestion.

Reviewer 3 Report

The manuscript entitled „ CHST7 methylation status related to the proliferation and dif-2 ferentiation of pituitary adenomas” presents results of the methylation status of CHST7 gene in the context of PAs and cell activity.

General comments.

The work describes the analyzes of various types of molecular data. The experiments themselves and the method of verification of the hypotheses are correct, but the description of the work itself is in many places unclear and very unscientific.

The authors describe mechanically what they have done, and do not show what is important from the point of view of the biological process they are trying to portray.

The text is written in a not very interesting way, it requires a very significant linguistic correction and a cause-and-effect description.

The work should undergo careful internal correction in the team and not be presented to external reviewers.

The Materials and Methods section is best described, while the results are the worst.

Please pay attention: often the elements/results that are shown in figures are also described in the text. Please do not repeat elements which can be read from a figure in the text.

Detailed comments:

Examples of unclear sentences: lines 21, 24, 43, 170, 171, 177, 178, 187, 189, 196, 216, 217, 233, 295, 297

Keywords should be in the alphabetic order.

Write gene symbols in italic CHST7 when referring to gene and not italic when referring to protein.

Line 147 – for the constructed model did you compute the BIC or Akaike?

I did not find information about correction for multiple testing e.g., results in tables. Please include correction for multiple testing.

Line 160 – you write that “there was no statistical difference”, but in line 161 given P=0.005.

Why you use capital symbol of p-value?

Please report degrees of freedom in statistical test results.

Why did you decide to make a cut-off point at 50% for hypermethylated cases? Please explain.

You present three classes: hypermethylated, hypomethylated and methylated. In my opinion the third class is misleading. A cytosine is methylated or not. You have information from a pool of cells and that is why methylation level might be between 0% and 100%. Please change the name of the third class e.g., middle/medium?

Line 185 – DEGs is from Differentially Expressed Genes and not “Differential genes (DEGs)…”

Line 194 – “Go” change to “GO”

Line 202- double dots

Line 203-211 – please re-write, confusing now. Paragraph is written with poor English and additionally you refer here to position in literature which is more confusing – which results are yours and which provided by other authors – please be specific here.

Figure 1D – presentation of this result is unclear.

Figure 2A – please use other type of results presentation – there are many informative ways of multiple correlation presentation.

Figure 2B – what is in columns? Please introduce this result in more informative way.

Lines 216-228 – this section is a good example showing that sometimes you describe everything you did in too details and you lose the ability to show what is important – home take message!

Due to numerous ambiguities in the text, and especially in the results, the unequivocal assessment of the section: "Discussion, is now difficult to assess.

Author Response

Detailed comments:

Examples of unclear sentences: lines 21, 24, 43, 170, 171, 177, 178, 187, 189, 196, 216, 217, 233, 295, 297

Keywords should be in the alphabetic order.

Write gene symbols in italic CHST7 when referring to gene and not italic when referring to protein.

Response:

Thanks for the question. We have re-edited the language according to the native editor.

Line 147 – for the constructed model did you compute the BIC or Akaike?

I did not find information about correction for multiple testing e.g., results in tables. Please include correction for multiple testing.

Response:

Thanks for the question. As we known, tumor size was usually not normal distribution, and we used the Mann-Whitney U test for tumor size in this study. There was no statistical difference based on CHST7 mRNA expression and methylation status. In this study, we focused on the cell differentiation not cell proliferation, hence, multiple testing was missing.

Line 160 – you write that “there was no statistical difference”, but in line 161 given P=0.005.

Why you use capital symbol of p-value?

Response:

Thanks for the question. They are writing errors, and we have correcting those in revised manuscript.

Please report degrees of freedom in statistical test results.

Why did you decide to make a cut-off point at 50% for hypermethylated cases? Please explain.

Response:

Thanks for the question. In our study, we found the CHSTY methylation levels in normal pituitary specimens (n=4) were 10.8±0.7%. and the criteria was: hypomethylation:≦ a 2 fold levels; methylation: 2-5 fold levels; hypermethylaiton > 5 fold levels.

You present three classes: hypermethylated, hypomethylated and methylated. In my opinion the third class is misleading. A cytosine is methylated or not. You have information from a pool of cells and that is why methylation level might be between 0% and 100%. Please change the name of the third class e.g., middle/medium?

Response:

Thanks for the question. We have adopted the reviewer’s suggestion in the revised manuscript.

Line 185 – DEGs is from Differentially Expressed Genes and not “Differential genes (DEGs)…”

Line 194 – “Go” change to “GO”

Line 202- double dots

Line 203-211 – please re-write, confusing now. Paragraph is written with poor English and additionally you refer here to position in literature which is more confusing – which results are yours and which provided by other authors – please be specific here.

Figure 1D – presentation of this result is unclear.

Response:

Thanks for the question. We have adopted the reviewer’s suggestion and re-edited the sentences in the revised manuscript.

Figure 2A – please use other type of results presentation – there are many informative ways of multiple correlation presentation.

Figure 2B – what is in columns? Please introduce this result in more informative way.

Lines 216-228 – this section is a good example showing that sometimes you describe everything you did in too details and you lose the ability to show what is important – home take message!

Due to numerous ambiguities in the text, and especially in the results, the unequivocal assessment of the section: "Discussion, is now difficult to assess.

Response:

Thanks for the question. We have adopted the reviewer’s suggestion and re-edited the sentences in the revised manuscript.

Round 2

Reviewer 1 Report

This manuscriptcript is a revised version of a paper concerning CHST7 methylation status in pituitary adenoma. While some of the reviewer’s comments have been addressed, there are still some flaws that should be fixed.

Figure 2 remained unchanged, soi t is still unclear. The legends in the particular graphs shown in Fig. 2A are to small; the particular bands (columns of bands) in Fig. 2B should be described (it is not enough to include the description in the leged). Furthermore, according to the newly provided legend, there are only tumor samples included in western blot, while non-tumor samples are absent; why those particular 8 tumor samples were shown, not the other? The sentence in lines 235-238 concerning Figure 2B is controversial: how were the correlation coefficients of particular proteins calculated? Please provide a detailed description of this calculation (how mane samples were taken into account? How were the samples divided into groups? What kind of data were actually used in this calculation?)

According to the curent version of the manuscript, non-tumor samples were not used in the western blot analysis; if so, in which part of the study were they used?  

The subtitle 3.6. is still unclear, as it suggests mechanistic insight which is not provided in this part of the text.

Author Response

Question 1: Figure 2 remained unchanged, soi t is still unclear. The legends in the particular graphs shown in Fig. 2A are to small; the particular bands (columns of bands) in Fig. 2B should be described (it is not enough to include the description in the leged). Furthermore, according to the newly provided legend, there are only tumor samples included in western blot, while non-tumor samples are absent; why those particular 8 tumor samples were shown, not the other? The sentence in lines 235-238 concerning Figure 2B is controversial: how were the correlation coefficients of particular proteins calculated? Please provide a detailed description of this calculation (how mane samples were taken into account? How were the samples divided into groups? What kind of data were actually used in this calculation?)

Response:

Thanks for the question. We have broken Figure 2 into Figure 2/3 and Supplementary Figure1/2 to improve the resolution.

We have added the description of the western blot experiment. In this study, we focused on the relationship of CHST7 and POU1F1/DLL3/EIF2AK3 in the same specimen according to the correlation analysis rather than groups, so that, we random chose 8 specimens. In fact, the fresh specimens in liquid nitrogen were far less than paraffin specimens in the most central.

We defined the gray value of patient 1 in Wb experiment as 1, and transformed the gray value of 8 patients into relative value in the correlation analysis.

Question 2: According to the current version of the manuscript, non-tumor samples were not used in the western blot analysis; if so, in which part of the study were they used ?  

Response:

Thanks for the question. Normal pituitary gland specimens were rare, and we only used them in methylation micro-arrays.

Question 3: The subtitle 3.6. is still unclear, as it suggests mechanistic insight which is not provided in this part of the text.

Response:

Thanks for the question. We have deleted the ambiguity word according to the suggestion as following:

Inhibin βE (INHBE) mRNA expression was positively correlated with CHST7.

Reviewer 2 Report

The authors answered to the different points of the revision satisfactorily.

Author Response

Thanks for the comments of the reviewer. 

Reviewer 3 Report

The authors have provided sufficient changes following the suggestions pointed to the earlier version of the manuscript. The current version fulfils the requirements.

Author Response

(The authors gave the same response as above.)

Round 3

Reviewer 1 Report

All the concerns have been addressed. Thank you for your efforts.

Author Response

(The authors gave the same response as above.)
